# *P-x,y* Equilibrium Data of the Binary Systems of 2-Propanol, 1-Butanol and 2-Butanol with Carbon Dioxide at 313.15 K and 333.15 K

**DOI:** 10.3390/molecules27238352

**Published:** 2022-11-30

**Authors:** Dragana Borjan, Maša Knez Marevci, Željko Knez

**Affiliations:** 1Laboratory for Separation Processes and Product Design, Faculty of Chemistry and Chemical Engineering, University of Maribor, 2000 Maribor, Slovenia; 2Laboratory for Chemistry, Faculty of Medicine, University of Maribor, 2000 Maribor, Slovenia

**Keywords:** high-pressure phase equilibrium, carbon dioxide, 2-propanol, 1-butanol, 2-butanol

## Abstract

The ability to predict the behaviour of high-pressure mixtures of carbon dioxide and alcohol is important for industrial purposes. The equilibrium composition of three binary carbon dioxide-alcohol systems was measured at temperatures of 313.15 K and 333.15 K and at pressures of up to 100 bar for carbon dioxide-2-propanol, up to 160 bar for carbon dioxide-1-butanol and up to 150 bar for carbon dioxide-2-butanol. Different equilibrium compositions of carbon dioxide in alcohols were observed despite their similar molecular weight (*M*_2-propanol_ = 60.100 g mol^−1^, *M*_1-butanol_ = 74.121 g mol^−1^ and *M*_2-butanol_ = 74.122 g mol^−1^) and place in the functional hydroxyl group (first or second carbon molecule). It is assumed that the differences in the phase equilibria are due to different vapor pressures, polarities and solute-solute interactions.

## 1. Introduction

Supercritical mixtures of carbon dioxide and alcohols are often encountered in natural gas, oil and petroleum industries, and these mixtures have been and will be studied extensively in the coming years [1,2,3]. Experimental vapour-liquid equilibria (VLE) data at elevated pressures provide important information to chemical engineers for process development, and is a requisite for many studies of the thermodynamics of the binary systems [4]. Given the complex nature of the molecular interactions in supercritical mixtures, the experimental work is an integral part of these studies. Critical point data obtained from these experiments enable the design and operation of the separation equipment and processes.

The increasing industrial and academic interest in high- and low-pressure data for gas-alcohol systems is confirmed by the large number of publications in this field [5,6,7,8,9,10]. Within this literature, sub- or supercritical carbon dioxide-methanol/ethanol systems have been studied widely [11,12,13,14,15], whereas sub- or supercritical carbon dioxide-other alcohol systems have attracted partial interest [16,17,18]. Joung et al. (2001) measured the high-pressure VLE data for binary carbon dioxide-alcohol systems (methanol, ethanol, 2-methoxyethanol and 2-ethoxyethanol) at temperatures from 313.15 K to 345.15 K and a pressure range from 6.0 bar to 124.0 bar. The obtained data were correlated by the Peng–Robinson and the multi-fluid non-random lattice fluid hydrogen-bonding equation of state (EoS) [16]. Fourie et al. (2008) studied the phase equilibria behaviour of four 8-carbon alcohols with supercritical carbon dioxide between the temperatures of 308.15 K and 348.15 K and pressures from 68.4 bar to 180 bar, to determine the effect of the hydroxyl group position on the alcohol solubility. Experimental bubble and dew point data were generated on a high-pressure phase equilibrium cell for the systems 1-octanol, 2-octanol, 3-octanol and 4-octanol in supercritical carbon dioxide between 308.15 K and 348.15 K. They determined a difference in the phase equilibria, which was a result of a difference in polarity [17]. The same group of authors continued researching the phase equilibrium of the supercritical carbon dioxide-alcohol systems under the same working conditions, and the results showed that the position, size and quantity of the side chains have a significant effect on the phase behaviour by changing the shape of the molecule and the effect of the hydroxyl group on the polarity of the molecule [18].

The low molecular mass alcohols are often used to adjust the polarity of supercritical carbon dioxide in extraction processes; and are also used as a cosolvent in supercritical fluid chromatography [19]. They are among the most important candidates for the application of extraction with supercritical fluids [20]. Hence, it is necessary to know the phase behavior and the entrainer effect of alcohol. Furthermore, high-pressure phase equilibrium data can support the compositions of coexisting phases, partition coefficients, mutual solubilities, and separation factors, which are essential in supercritical fluid applications. Accordingly, the phase equilibrium data are necessary for the fundamental calculations in the practical design of the supercritical processes.

Carbon dioxide has emerged as the most commonly used supercritical fluid for these processes, due to its excellent thermodynamic and transport properties and low price. Carbon dioxide has a low critical temperature of 304.25 K and a critical pressure of 73.8 bar [21].

To ensure the accuracy and reliability of the measurements, the experimental data obtained in the current study were compared with those in the available literature. Lim et al. (2007) reported isothermal high-pressure VLE data for a binary mixture of carbon dioxide-2-propanol at various temperatures from 323.15 K to 353.15 K, and pressures of up to 107.7 bar, and correlated the results with the Peng–Robinson EoS combined with the Wong–Sandler mixing rule [22]. Seciuanu et al. (2003) reported the VLE data for a carbon dioxide-2-propanol system at 317.15 K and with pressures of up to 80.6 bar [23].

The current research work was focused on a systematic study to obtain the *P-x*,*y* equilibrium data for the binary systems containing carbon dioxide and alcohol, experimentally. These data have not yet been studied in detail and have not been published in the literature. The experiments were performed at two different temperatures (313.15 K and 333.15 K) and at pressures of up to 100 bar for 2-propanol, up to 160 bar for 1-butanol and up to 150 bar for 2-butanol. A variable-volume high-pressure optical view cell was used for all experiments.

## 2. Experimental

### 2.1. Materials

2-propanol was obtained from Honeywell with a 99.8 vol. % purity (CAS Reg. No. 33539-2.5L). 1-butanol (CAS Reg. No. 71-36-3) and 2-butanol (CAS Reg. No. 78-92-2) were obtained from Sigma Aldrich (Darmstadt, Germany) with the purity of 99.0 vol. %. The carbon dioxide (CAS Reg. No. 124-38-9) with a purity of 99.99 vol. % was obtained from MESSER (MG-Ruše, Slovenia). All of the chemicals were used as received without further purification.

### 2.2. Apparatus and Procedure

The phase behavior of alcohols (2-propanol, 1-butanol and 2-butanol) in the presence of carbon dioxide, was investigated using a high-pressure variable-volume optical view cell with a sapphire glass on the front and back (NWA GMBh, Lorrach, Germany), as shown in Figure 1.

The apparatus was designed for a pressure of 750 bar and temperature of 473.15 K, with a variable volume that can range between 30 mL and 60 mL by means of a hydraulic sapphire piston. The hydraulic sapphire piston was connected to a hydraulic pressurisation system, the position of which was determined by measuring the electric resistance through a cable sealed to the hydraulic sapphire piston and connected to an ohmmeter (mod. 2210, MISCO-Systemax Europe Ltd., Wellingborough, UK). The pressure inside the cell was increased using an inlet gas flow and a high-pressure pump (NWA PM-101, Lorrach, Germany). The pressure was measured using an electronic pressure gauge (WIKA Alexander Wiegand GmbH & Co. KG, Klingenberg, Germany) with an uncertainty of ± 0.1 bar. In addition to the two sapphire windows, the cell included a thermocouple for the temperature monitoring (accurate ± 0.5 K) and with two valves for loading and discharging the gas. The cell contained a blade turbine stirrer for the mixing phases and two 200 W electric heaters (mod. Firerod, WATLOW, St. Louis, MO, USA) inserted in the stainless-steel coat of the cell [25,26].

The addition of alcohol into the cell was performed at ambient conditions using a volumetric method. At the beginning of measurements, the high-pressure optical view cell volume with a hydraulic sapphire piston at the start position was 60 mL. The volume of added alcohol was 30 mL. Consequently, the initial composition has been assumed as 50 vol. %. The cell volume had been varied only by a hydraulic sapphire piston, as shown in Figure 2. The stirrer does not affect the reduction of volume. Furthermore, the possible moving positions of the hydraulic sapphire piston are indicated by an arrow and it can be seen that the hydraulic sapphire piston does not have contact with the stirrer.

The binary mixture was heated until the operating temperature was reached, and the system was pressurised to the desired pressure. In the sampling procedure the temperature was kept constant during all measurements. The system with 2-propanol was mixed with a blade-turbine stirrer operated at 200–300 rpm for 1 h and with butanol isomers at 300–400 rpm for 4 h. During the separation time of 1 h for the system with 2-propanol and 4 h for the systems with butanol isomers, the system settled down and the phases were separated, and, thus, the equilibrium was reached. To keep the pressure constant during the sample procedure, the hydraulic sapphire piston reduces the volume of the cell. Both phases, the liquid rich phase as well as the gas rich phase, were sampled into glass traps, which were cooled to −10 °C. The mass of the sample in the trap was determined gravimetrically (accuracy ± 0.0001 g) after equilibration at ambient conditions. Furthermore, the sampling trap was exposed to the ultrasound bath to release the solubilised gas. The volume of the gas phase released during sampling was measured and its mass was calculated the using ideal gas law. At least three experiments were performed for each measured point, and the data obtained were similar (to within ±5 bar), and thus the results were comparable and repeatable (deviation was ±2.30% for 2-propanol, ±1.16% for 1-butanol and ±1.48% for 2-butanol).

## 3. Results and Discussion

The phase transition from a two- to a one-phase system for the carbon dioxide-2-propanol system is detected at 70 bar for 313.15 K and at above 100 bar for 333.15 K; for the carbon dioxide-1-butanol system, it is detected at above 157 bar for 313.15 K and at above 122 bar for 333.15 K; for the carbon dioxide-2-butanol system, the phase transition is detected at above 74 bar for 313.15 K and at above 142 bar for 333.15 K.

In addition, the temperature dependence of the equilibrium composition can vary depending on the solvent power of the supercritical fluid. In principle, as the pressure increases at a constant temperature, the density of the solvent increases. The distance between the solvent molecules decreases, thereby increasing the specific interactions between the solute and solvent molecules. The next parameter that affects the solubility is the temperature. This affects the vapor pressure of the solute and the density of the solvent. At low pressures, even with a slight increase in temperature, the density of the solvent drops rapidly. The effect of the density dominates in this range and therefore the solubility decreases with the increasing temperature. At higher pressures, the density of the solvent depends only slightly on the temperature, so the solubility will increase due to the increase in the vapor pressure of the solute.

It is difficult to predict which of the influences will dominate in a particular case, so it is necessary to determine the solubilities of the substances depending on the pressure and temperature.

The results obtained for the carbon dioxide-2-propanol system (Figure 3) show that the equilibrium composition of the carbon dioxide in 2-propanol increases with the increasing pressure at a constant temperature. Additionally, the equilibrium compositions at 313.15 K are lower than at 333.15 K for similar pressures; therefore, the equilibrium composition also increases with the temperature. These conclusions are in line with those of many previous researchers [23,27,28].

Lim et al. (2007) measured isothermal high-pressure VLE data for the binary mixtures of carbon dioxide-2-propanol at various temperatures from 323.15 K to 353.15 K. The vapour and liquid compositions and pressures were measured in a circulation-type apparatus. The equilibrium composition results obtained in this work are in quite good agreement with their literature data (Figure 3). For instance, at a temperature of 333.15 K and pressure of 17.5 bar obtained in this work, the equilibrium composition of carbon dioxide in 2-propanol is 0.1041 mole fractions, while at the same temperature and similar pressure (20.1 bar), the equilibrium composition of carbon dioxide in 2-propanol is 0.1026 mole fractions, according to Lim et al. (2007) [22]. Additionally, Secuianu et al. (2003) measured the high-pressure VLE for a binary carbon dioxide-2-propanol system, but at a temperature of 317.15 K and at pressures of up to 80.6 bar [23]. According to the different working conditions, these results are not compared to the results obtained in this work. However, the shape of the trend lines in both studies is similar.

Furthermore, Secuianu et al. (2008) measured the isothermal (P, T, *x*, *y*) data for the similar binary system carbon dioxide-1-propanol at temperatures from 293.15 K to 353.15 K and at pressure from 6.1 bar to 126.4 bar. Their results are comparable with the results obtained in this work at both investigated temperatures. At a temperature of 313.15 K and a pressure of 14.1 bar, the equilibrium composition of carbon dioxide in 2-propanol is 0.1008 mole fractions, and, according to the literature, at a pressure of 15.5 bar, the equilibrium composition of carbon dioxide in 1-propanol is 0.0890 mole fractions. At a temperature of 333.15 K, the deviation is slightly larger. However, the shape of the trend line is still the same: at a pressure of 93.6 bar, the equilibrium composition of carbon dioxide in 2-propanol is 0.6987 mole fractions, and at a pressure of 93.5 bar, the equilibrium composition of carbon dioxide in 1-propanol is 0.5285 mole fractions [27].

The results obtained for the carbon dioxide-1-butanol system (Figure 4) show that the equilibrium fraction of carbon dioxide in 1-butanol increases with the increasing pressure at a constant temperature. Additionally, the equilibrium fractions at 313.15 K are higher than at 333.15 K for similar pressures; therefore, the equilibrium fraction decreases with the increasing temperature. These conclusions are in line with the previously reported data [29,30].

The results obtained for the carbon dioxide-2-butanol system (Figure 5) show that the equilibrium fraction of carbon dioxide in 2-butanol increases with the increasing pressure at a constant temperature.

Additionally, the equilibrium fractions at 333.15 K are lower than at 313.15 K for similar pressures; therefore, the equilibrium fraction decreases with the temperature.

Unfortunately, there are no data available in the literature on the equilibrium fraction of carbon dioxide in 2-butanol that can be used for comparison.

The obtained conclusions are in line with those found by other authors [17]. The polarity introduced by the primary hydroxyl group reduces the solubility of 1-butanol, resulting in higher phase transition pressures. Shifting the hydroxyl group from the first to second carbon causes a large decrease in the polarity and increase in the solubility.

Further movements toward the molecule centre result in much smaller polarity reductions and solubility increases, producing phase boundaries that coincide or differ minimally.

As VLE data and accurate thermodynamic models for mixtures are basic requirements for the design, simulation, operation, and optimization of the industrial processes, several authors have investigated the mixtures involving carbon dioxide and alcohols in the sense of interest for a range of industrial ventures. There have also been some papers published on various modelling studies on this topic. Nguyen Huynh et al. (2014) applied a polar version of the group contribution PC-SAFT (perturbed chain-statistical associating fluid theory) equation of the state combined with a method for the correlation/prediction of the binary interaction parameters *k_ij_* to model the vapour-liquid, liquid-liquid and vapour-liquid-liquid phase equilibria of carbon dioxide-alkanol mixtures, simultaneously [31]. Raeissi et al. (2015) measured the bubble points of the binary mixtures of carbon dioxide and secondary butyl alcohol, using a synthetic method. The measurements covered a carbon dioxide molar concentration range from 0.10% to 0.57% and temperatures from 293 K to 370 K, with pressures reaching up to 110 bar. The experimental data were modelled by the cubic plus association equation of state, as well as the more simple Soave–Redlich–Kwong equation of state [32]. Oliveira et al. (2011) investigated how the cubic plus association equation of state can be used for an adequate description of the vapour-liquid equilibrium of an extensive series of carbon dioxide binary systems containing n-alkanes, n-alcohols, esters and n-acids, in a broad range of temperatures and pressures [33]. Smith et al. (2021) studied a modelling approach for the binary mixtures of carbon dioxide or benzene with n-alkanes and 1-alkanols and compared the performance of the cubic plus association equation of state extended with the theory of the quadrupolar molecules of Gross [34].

Moreover, some authors studied similar systems but for multicomponent mixtures. Tsivintzelis and Kontogeorgis (2015) evaluated the performance of the cubic plus association equation of state for the ternary and multicomponent carbon dioxide mixtures containing alcohols (methanol, ethanol or propanol), water and hydrocarbons [35].

The present manuscript represents the novel data on the binary systems of 2-propanol; 1-butanol and 2-butanol with carbon dioxide which has not yet been published elsewhere. Therefore, the measured data cannot be compared to those in the literature and the previously inconclusive differences among the data of the various groups is now aided with a new dataset, which coincides well with two of the literature studies on similar systems, perhaps making the choice among the various datasets easier for the future.

## 4. Conclusions

Binary systems of carbon dioxide and three different alcohols (2-propanol, 1-butanol and 2-butanol) were investigated, and the equilibrium compositions were determined at temperatures of 313.15 K and 333.15 K and pressures of up to 100 bar for 2-propanol, up to 160 bar for 1-butanol and up to 150 bar for 2-butanol, using a variable-volume high-pressure optical view cell. The results are novel, and attained by a simple, economic method which is also adjustable for various multi-compound mixtures at elevated pressures.

The accurate prediction of the equilibrium, thermodynamic and mass transfer data is fundamental in various engineering and industrial operations to design processes involving mass transfer (e.g., conventional and supercritical extractions, multiphase chemical reactions, distillation, carbon sequestration, membrane separation processes, absorption and adsorption). These data, measured at ambient conditions, can be found in the literature for numerous binary and ternary systems, since the literature that reports the data on the systems with supercritical fluids is still relatively scarce.

The results obtained for all three investigated carbon dioxide-alcohol systems show that the equilibrium composition of carbon dioxide in alcohol increases with the increasing pressure at a constant temperature. Additionally, equilibrium compositions at 313.15 K are lower than at 333.15 K for similar pressures; therefore, the equilibrium composition also increases with the temperature. The obtained conclusions are in line with those found by other authors.

## Figures and Tables

**Figure 1 molecules-27-08352-f001:**
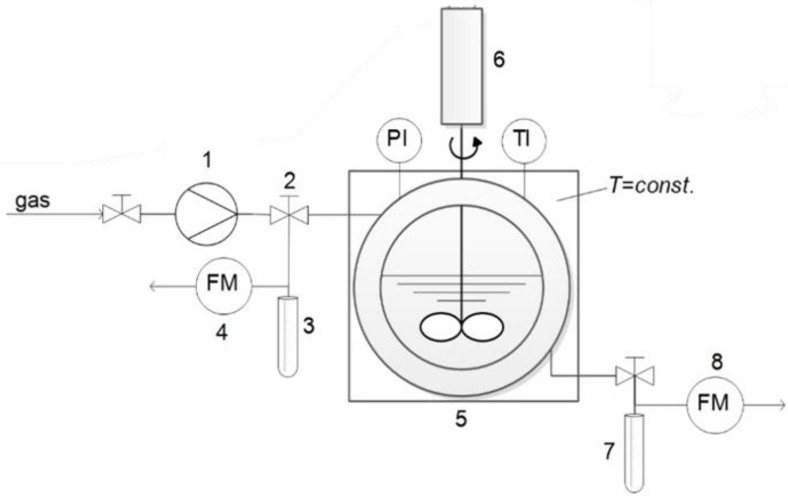
A scheme of a variable-volume high-pressure optical view cell for the determination of the solubility of argon-butanol isomer systems: 1-driven gas booster; 2-argon supply valve and valve for gas-rich phase sampling; 3-glass trap for sampling; 4-gas flow meter; 5-optical view cell; 6-propeller mixer; 7-glass trap for sampling; 8-gas flow meter [24].

**Figure 2 molecules-27-08352-f002:**
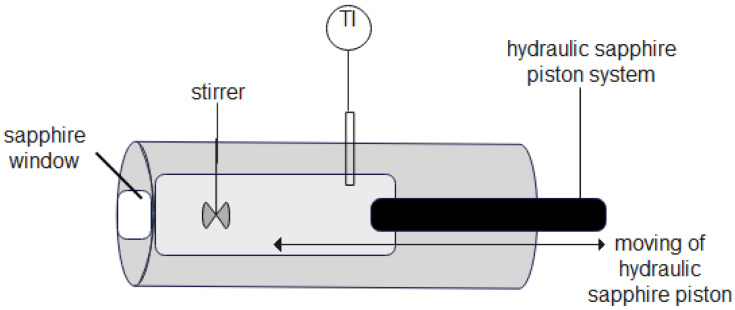
Side view of a scheme of a variable-volume high-pressure optical view cell with a hydraulic sapphire piston system [24].

**Figure 3 molecules-27-08352-f003:**
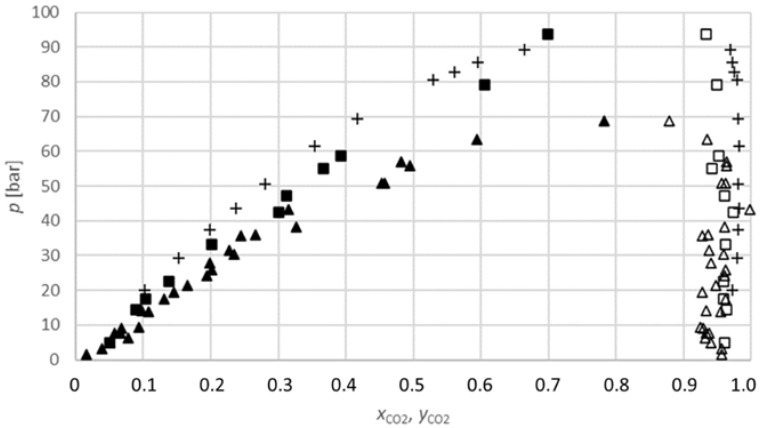
*P-x,y* diagram for the phase equilibria for the CO_2_-2-propanol binary system (*x*_CO2_, *y*_CO2_: CO_2_ mole fraction): ▲, measured data at 313.15 K (lower phase); ■, at 333.15 K (lower phase); ∆, at 313.15 K (upper phase); □, at 333.15 K (upper phase); +, Lim et al. (2007) at 333.15 K [22].

**Figure 4 molecules-27-08352-f004:**
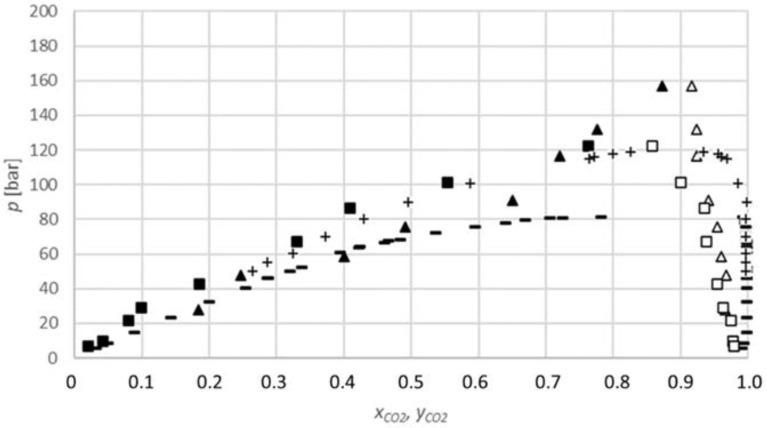
*P-x,y* diagram for the phase equilibria for the CO_2_-1-butanol binary system (*x*_CO2_, *y*_CO2_: CO_2_ mole fraction): ▲, measured data at 313.15 K (lower phase); ∆, at 313.15 K (upper phase); ■, at 333.15 K (lower phase); □, at 333.15 K (upper phase); −, Secuianu et al. (2004) at 313.15 K [29]; +, Chen et al. (2002) [30].

**Figure 5 molecules-27-08352-f005:**
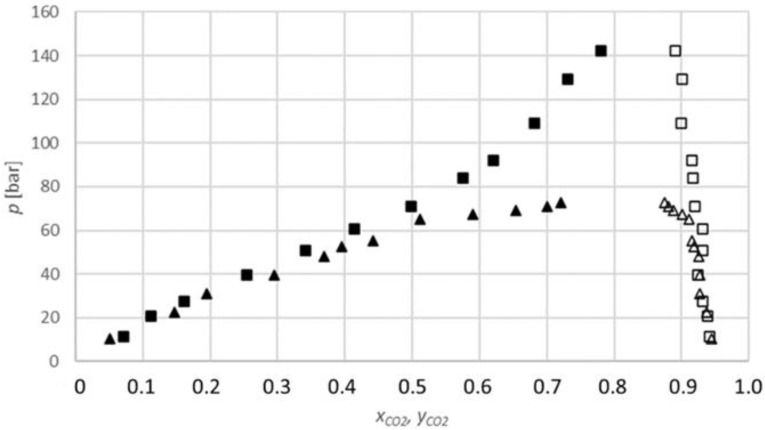
*P-x,y* diagram for the phase equilibria for the CO_2_-2-butanol binary system (*x*_CO2_, *y*_CO2_: CO_2_ mole fraction): ▲, measured data at 313.15 K (lower phase); ∆, at 313.15 K (upper phase); ■, at 333.15 K (lower phase); □, at 333.15 K (upper phase).

## Data Availability

Not applicable.

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
