# Peer review of "P-x,y Equilibrium Data of the Binary Systems of 2-Propanol, 1-Butanol and 2-Butanol with Carbon Dioxide at 313.15 K and 333.15 K"

_molecules, 2022, doi:10.3390/molecules27238352_

Round 1
Reviewer 1 Report
The work presented by Borjan and co-workers presents experimental data of the pressure-composition curve at two temperatures (313.15 and 333.15 K) for several mixtures of carbon dioxide - alcohol, namely 2-propanol, 1-butanol and 2-butanol. The work presents data which fits well with previous data, specially for 1-propanol.
The work is in general well structured and written. The main objective is clear and the results are presented in a transparent manner. However, the work requires several clarifications prior to acceptance.
In order to improve the impact of the work, I encourage the authors to pay attention to the following issues:
1) In the introduction, the authors describe the interest on supercritical mixtures of carbon dioxide - alcohol as a field of current and continuous interest. However, among references 1-15 only 3 are really recent (refs. 1, 5 and 14), the remaining references came from around 20 years ago. I would encourage the authors to describe briefly the industrial processes in which the knowledge of the high pressure behavior of CO2-alcohol mixtures is indeed required.
2) In section 2.2, It is not clear for me the physical principles used in the mass calculation. A brief comment about the procedure to determine the molar concentration (reported later in section 3) would be very useful.
3) The results presented in Fig. 4 only share qualitative trends with previous data, while the measured data show important quantitative differences with previous experiments of similar systems. I would appreciate additional discussion about the possible reasons for such great differences, particularly around the maximum. It is especially intriguing the shape of the 313.15 K-data, they look very sharp compared with previous results. Could the authors comment about the reliability of that result?
4) In Fig. 5, a very sharp shape is observed again for the CO2- 2 butanol binary system. Can the authors discuss this issue?
Author Response
Reviewer 1
The work presented by Borjan and co-workers presents experimental data of the pressure-composition curve at two temperatures (313.15 and 333.15 K) for several mixtures of carbon dioxide - alcohol, namely 2-propanol, 1-butanol and 2-butanol. The work presents data which fits well with previous data, specially for 1-propanol.
The work is in general well structured and written. The main objective is clear and the results are presented in a transparent manner. However, the work requires several clarifications prior to acceptance.
In order to improve the impact of the work, I encourage the authors to pay attention to the following issues:
1) In the introduction, the authors describe the interest on supercritical mixtures of carbon dioxide - alcohol as a field of current and continuous interest. However, among references 1-15 only 3 are really recent (refs. 1, 5 and 14), the remaining references came from around 20 years ago. I would encourage the authors to describe briefly the industrial processes in which the knowledge of the high pressure behavior of CO2-alcohol mixtures is indeed required.
Response: It has been changed, please check the Manuscript file.
2) In section 2.2, It is not clear for me the physical principles used in the mass calculation. A brief comment about the procedure to determine the molar concentration (reported later in section 3) would be very useful.
Response: It was calculated using the ideal gas law – it has been added in section 2.2, line 125.
3) The results presented in Fig. 4 only share qualitative trends with previous data, while the measured data show important quantitative differences with previous experiments of similar systems. I would appreciate additional discussion about the possible reasons for such great differences, particularly around the maximum. It is especially intriguing the shape of the 313.15 K-data, they look very sharp compared with previous results. Could the authors comment about the reliability of that result?
Response: The method used has been validated on the systems with known data (solubility has already been published in scientific literature data). Additionally, obtained experimental data at 333.15 K are comparable to literature data. On the other hand, the phase transition from a two-phases to a one-phase system at 313.15 K was not reached in the literature and it is probably the reason for the different sharp, especially as literature and experimental points are close at pressures up to 50 bars.
4) In Fig. 5, a very sharp shape is observed again for the CO2- 2 butanol binary system. Can the authors discuss this issue?
Response: It had been already explained, please check the Manuscript file, lines 135-145.
Reviewer 2 Report
Review of Manuscript ID: molecules-2014486
Title: “P-x,y Equilibrium Data of the Binary Systems of 2-Propanol, 1-Butanol and 2-Butanol with Carbon Dioxide at 313.15 K and 333.15 K”
Authors: “Borjan et al.”
The current paper presents a series of experimental measurements for 3 binary systems that contain CO2 and alcohol. Such studies are useful since there is usually scarcity of experimental data.
I would recommend publication of the manuscript after a number of issues are addressed in an adequate manner.
In particular:
1) In order for the paper to be useful to other researchers as well, the measured values should also be reported in Tables that also include the errorbars for all the 3 systems considered. Such Tables could be included as a Supplementary Material file.
2) In order to improve the clarity of Figures: (i) the size of the figures need to be increased and (ii) the errorbars in figure 3 should be removed and report them in the Tables (see previous comment).
3) The authors state in lines 154-156: “It is not possible to predict which of the influences will dominate in a particular case, so it is necessary to experimentally determine the solubilities of substances depending on pressure and temperature.” I believe that the specific statement is not accurate. In particular, a number of modeling studies that use Equations of State (e.g., CPA, SAFT-type, etc.) have been reported that provide a very good description of systems such as the ones examined in the current study. The following studies are some characteristic typical examples (many more can be found) : (i) J. Supercritical Fluids, 95, 146-157, 2014; (ii)J. Solution Chem., 44, 1555-1567, 2015; (iii) J. Supercritical Fluids, 55, 876-892, 2011; (iv) J. Supercritical Fluids, 104, 29-39, 2015; (v) Fluid Phase Equilib., 528, 112848, 2021. The aforementioned studies include some additional experimental data that the authors have not considered. Therefore: (a) it would be useful to include comparison with the additional experimental data, and (b) include a brief discussion of modeling studies.
Author Response
Reviewer 2
Title: “P-x,y Equilibrium Data of the Binary Systems of 2-Propanol, 1-Butanol and 2-Butanol with Carbon Dioxide at 313.15 K and 333.15 K”
Authors: “Borjan et al.”
The current paper presents a series of experimental measurements for 3 binary systems that contain CO2 and alcohol. Such studies are useful since there is usually scarcity of experimental data.
I would recommend publication of the manuscript after a number of issues are addressed in an adequate manner.
In particular:
1) In order for the paper to be useful to other researchers as well, the measured values should also be reported in Tables that also include the errorbars for all the 3 systems considered. Such Tables could be included as a Supplementary Material file.
Response: All data with standard deviations for three measurements were already provided in the Supplementary Material file.
2) In order to improve the clarity of Figures: (i) the size of the figures need to be increased and (ii) the errorbars in figure 3 should be removed and report them in the Tables (see previous comment).
Response: Figures had been prepared according to journal’s template, but as suggested they have been increased. Error bars have been removed. Please check the Manuscript as well as the Supplementary Material files.
3) The authors state in lines 154-156: “It is not possible to predict which of the influences will dominate in a particular case, so it is necessary to experimentally determine the solubilities of substances depending on pressure and temperature.” I believe that the specific statement is not accurate. In particular, a number of modeling studies that use Equations of State (e.g., CPA, SAFT-type, etc.) have been reported that provide a very good description of systems such as the ones examined in the current study. The following studies are some characteristic typical examples (many more can be found) : (i) J. Supercritical Fluids, 95, 146-157, 2014; (ii)J. Solution Chem., 44, 1555-1567, 2015; (iii) J. Supercritical Fluids, 55, 876-892, 2011; (iv) J. Supercritical Fluids, 104, 29-39, 2015; (v) Fluid Phase Equilib., 528, 112848, 2021. The aforementioned studies include some additional experimental data that the authors have not considered. Therefore: (a) it would be useful to include comparison with the additional experimental data, and (b) include a brief discussion of modeling studies.
Response: Suggested articles have been included in a discussion, please check the Manuscript file, lines 203-222. Due to a different compounds studied and under different conditions than in our article, it was not possible to directly compare the data.